# The risk of dyslipidemia on PLHIV associated with different antiretroviral regimens in Huzhou

Yanan Wang[1,2,3☯], Zhongrong Yang[1☯], Jing Li[1], Zhenqian Wu[1], Xiaoqi Liu[1], Hui Wang[2,3], Yuxin Chen[2,3], Ziyi Wang[2,3], Zhaowei Tong[4], Xiaofeng Li[4], Feilin Ren[1], Meihua Jin[1]*, Guangyun Mao[2,3,5]*

1 Huzhou Center for Disease Control and Prevention, Huzhou, Zhejiang, China, 2 Division of Epidemiology and Health Statistics, Department of Preventive Medicine, School of Public Health & Management, Wenzhou Medical University, Wenzhou, Zhejiang, China, 3 Center on Evidence-Based Medicine & Clinical Epidemiological Research, School of Public Health & Management, Wenzhou Medical University, Wenzhou, Zhejiang, China, 4 Department of Infectious Diseases, Huzhou Central Hospital, Huzhou, Zhejiang, China, 5 National Clinical Research Center for Ocular Diseases, Wenzhou, Zhejiang, China

☯ These authors contributed equally to this work.
* huzhoujmh6821@163.com (MJ); mgy@wmu.edu.cn (GM)

**Data Availability Statement:** Data cannot be shared publicly because of the data involves information of people living with HIV, which is a sensitive topic in Chinese culture. Data are available from the Ethics Committee of Huzhou center for

## Abstract

### Background

Dyslipidemia is increasingly common in people living with HIV (PLHIV), thereby increasing the risk of cardiovascular events and diminishing the quality of life for these individuals. The study of blood lipid metabolism of PLHIV has great clinical significance in predicting the risk of cardiovascular disease. Therefore, this study aims to examine the blood lipid metabolism status of HIV-infected patients in Huzhou before and after receiving highly active antiretroviral therapy (HAART) and to explore the impact of different HAART regimens on dyslipidemia.

### Method

PLHIV confirmed in Huzhou from June 2010 to June 2022 was included. The baseline characteristics and clinical data during the follow-up period were collected, including some blood lipid indicators (total cholesterol and triglycerides) and HAART regimens. A multivariate logistic regression model and the generalized estimating equation model were used to analyze the independent effects of treatment regimens on the risk of dyslipidemia.

### Result

The overall prevalence of dyslipidemia among PLHIV after HAART was 70.11%. PLHIV receiving lamivudine (3TC) + efavirenz (EFV) + zidovudine (AZT) had a higher prevalence of dyslipidemia compared to those receiving 3TC+EFV+tenofovir disoproxil fumarate (TDF). In a logistic analysis adjusted for important covariates such as BMI, age, diabetes status, etc., we found that the risks of dyslipidemia were higher with 3TC+EFV+AZT (dyslipidemia: odds ratio [OR] = 2.09, 95% confidence interval [CI]: 1.28–3.41; TG ≥1.7: OR = 2.40, 95%

disease control and prevention (contact via Zhongrong Yang with email: yzhr91@126.com) for researchers who meet the criteria for access to confidential data.

**Funding:** This work was supported by Medical and Health Research Project of Zhejiang Province (2022KY369 to JMH), Huzhou science and technology research plan project (2022GYB13 to LXQ), the Huzhou Medical Key Supporting Discipline (Epidemiology, to JMH), and the Key Laboratory of Emergency detection for Public Health of Huzhou (to JMH).

**Competing interests:** The authors have declared that no competing interests exist.

CI:1.50–3.84) than with 3TC+EFV+TDF. Furthermore, on PLHIV that was matched 1:1 by the HAART regimens, the results of the generalized estimation equation again showed that 3TC+EFV+AZT (TG $\geq$1.7: OR = 1.84, 95%CI: 1.10–3.07) is higher for the risk of marginal elevations of TG than 3TC+EFV+TDF.

## Conclusion

The prevalence of dyslipidemia varies according to different antiretroviral regimens. Using both horizontal and longitudinal data, we have repeatedly demonstrated that AZT has a more adverse effect on blood lipids than TDF from two perspectives. Therefore, we recommend caution in using the 3TC+EFV+AZT regimen for people at clinical risk of co-occurring cardiovascular disease.

## Introduction

The widespread use of HAART has led to an extended life expectancy for PLHIV. However, the burden of non-AIDS-related diseases, such as dyslipidemia, type 2 diabetes, and cardiovascular disease (CVD), steadily increasing annually [1]. CVD complications have become a significant cause of death among HIV-infected patients [2]. Previous studies have demonstrated that people newly diagnosed with HIV have a higher risk of cardiovascular disease compared to the general population [3, 4]. Following prolonged exposure to ART, the drugs are likely to further increase the risk of CVD by exacerbating mitochondrial toxicity accumulation, enhancing lipid biosynthesis function, and reducing the liver's capacity to clear lipids, among other mechanisms [5]. Furthermore, with exposure to different combinations of antiretroviral drugs, the risk may vary widely [6].

In developing countries, first-line HAART regimens typically comprises a combination of at least three drugs, primarily two nucleoside reverse transcriptase inhibitors (NRTI) and one non-nucleoside reverse transcriptase inhibitors (NNRTI) or integrase inhibitor. The selection of the appropriate drug combination is contingent on a range of factors, including HIV viral load, CD4 and CD8 counts, drug interactions and more [7]. At present, the preferred combination of free first-line treatment for adults in China is 3TC+EFV+TDF. Other free alternative first-line treatment regimens include 3TC+AZT+EFV, 3TC+AZT+nevirapine (NVP) and TDF+3TC/ NVP/emtricitabine (FTC). As an important modifiable risk factor for PLHIV atherosclerotic cardiovascular disease (ASCVD) [8], dyslipidemia also influences the selection and implementation of HAART.

However, the current research data on abnormal blood lipid metabolism in HIV-infected patients primarily originates from abroad. China's first-line ART program differs from those of developed countries, and the sociodemographic characteristics are also distinct from those of foreign nations. The existing domestic studies have a short follow-up period, a limited sample size, and the changes in blood lipid metabolism are not fully elucidated. This study used comprehensive clinical research data on HIV/AIDS patients in Huzhou from July 2005 to June 2022 to estimate the prevalence of dyslipidemia among PLHIV before and after long-term ART regimens. The objective was to assess the trend of dyslipidemia and evaluate the association between different HAART regimens and dyslipidemia in PLHIV. Early intervention for disorders of lipid metabolism can potentially prevent possible CVD events, thereby providing clinical decision-making guidance throughout the entire HIV management process.

## Materials and methods

### Study participants

This study was a retrospective cohort study involving 1,876 PLHIV who sought treatment from the China Disease Prevention and Control Information System (CDPCIS) in the Huzhou area, China. The inclusion criteria were as follows: over 18 years of age; having no received HAART; diagnosed or identified from July, 2005 to June, 2022; living in the Huzhou area containing temporary residents. The exclusion criteria were having no baseline and at least one follow-up measurements of blood lipids; presence of liver disease (baseline aminotransferases are elevated more than 3 times the upper limit or bilirubin levels are elevated more than 2.5 times above the upper limit) or chronic kidney disease (Serum creatinine levels exceed the upper limit by 1.5 times). Since the number of participants receiving other ART regimens except for 3TC+EFV +AZT and 3TC+EFV+TDF was small and scattered, these individuals were also excluded from the final analysis to facilitate the interpretation of the results. The data analyses of the study were conducted from July to December 2022. We first accessed the patient data in July 2022.

### Study design

The overview of the study workflow is shown in the supporting information of S1 Fig. In the primary analysis, we enrolled 532 participants, of which 391 were on an initial ART regimen of 3TC +EFV+TDF,while 141 were on 3TC+EFV+AZT at the baseline. These patients were guaranteed to have complete baseline data and at least one follow-up visit data. In the secondary analysis, we further selected 476 participants from the 532 PLHIV who did not change their HAART regimen at any follow-up. Due to the irregularfollow-up time for these patients, we fixed four time points at 1-year intervals (baseline, 12 months, 24 months, 36months, using a +/- 3 months window). To enhance the accuracy of the dynamic analysis of lipids, we retained 376 PLHIV who had at least two follow-up lipid data. Subsequently, PLHIV receiving two HAART regimens were matched by age, BMI, CD4 count, CD4/CD8 ratio, and baseline lipid level at a ratio of 1:1, using a propensity score matching approach.Ultimately, 77 patients were included in each of the 3TC+EFV+TDF and 3TC+EFV+AZT two regimens, and their longitudinal lipid data were analysed.

### Covariates

The baseline data of the participants, serving as covariates, were extracted from the CDPCIS, encompassingdemographic and clinicalcharacteristics, as well as laboratory results. The baseline was defined as the date when the participants first sought care for HAART at the local health system. The demographic and clinical data, collected through a face-to-face interview and physical examination, included age, gender, weight, height, initial HAART regimens, route of transmission, education level, WHO clinical stage, and others. BMI was calculated as the weight (kg) divided by the square of height (m2). Information on laboratory tests, strictly measured by local AIDS-designated treatment hospitals, included total cholesterol (TC), triglyceride (TG), CD4 cell count (CD4), CD8 cell count (CD8), serum hemoglobin, platelet (PLT), white blood cell (WBC), alanine aminotransferase (ALT), aspartate transaminase (AST), serum creatinine, total bilirubin (TBIL), and fasting plasma glucose (FPG). The diagnosis criterion fora history of diabetes mellitus in PLHIV wasa baseline FPG of $\geq$ 7.0 mmol/L or having received treatment for diabetes.

### Definition of dyslipidemia

Dyslipidemia was determined according to *the Guidelines for the Prevention and Treatment of Dyslipidemia in Adults in China (2016 Revision)*.Participants were categorized as having dyslipidemia if they exhibited one or more of the following conditions: TC $\geq$ 5.2mmol/L or

TG $\geq$ 1.7mmol/L or low-density lipoprotein cholesterol (LDL-C)$\geq$ 3.37mmol/L or high-density lipoprotein cholesterol (HDL-C) $\leq$ 1.04mmol/L. The survey lacked indicators for both HDL_C and LDL_C, therefore, dyslipidemia was considered to be present as long as there was a marginal elevation in lipid profiles during the follow-up period in order to avoid underestimation the outcomes.

### Ethical approval

The study was approved by the Research Ethics Committee of Huzhou Center for Disease Control and Prevention. Written informed consent was given by all participants. All procedures performed were carried out in accordance with the ethical standards laid down in the 1964 Declaration of Helsinki and its later amendments.

### Statistical analysis

We described continuous variables using mean ± standard deviation (SD) or median and interquartile range (IQR), and the Student $t$-test or Wilcoxon rank sum test was used to compare differences. Categorical variables were presented as proportions, and chi-square or Fisher's exact tests were used for their comparisons. The prevalence of dyslipidemia and its components in PLHIV with varying characteristics were caculated. Missing values were imputed using a 5-fold multiple imputation approach, and a sensitivity analysis was conducted on the comparison of pre- and post-imputation to validate the stability of the imputations (the supporting information of S1 Table).

We used two analytical methods to investigate how the probability of experiencing dyslipidemia depends on major HAART regimens. Firstly, a multivariate logistic regression model was employed to determine the statistical association of two, adjusting for potential confounders, including age, BMI, ALT, AST, PLT, CD4/CD8 and history of diabetes ($p$-value $<0.1$ in the univariate analysis) at baseline. Then, to address the concern that lipid profiles may change over time and the measurement at a single time point measurement may not be sufficient, we examined the association between lipid status over time and two HAART regimens that never change using a generalized estimating equation (GEE) model. Here, an exchange correlation structure was used as the GEE model's working correlation matrix.

All data management and statistical analyses were conducted using R Version 4.2.0 (Copyright© 2022 The R Foundation for Statistical Computing). All tests were two-side and P $\leq$ 0.05 was set as the significant level.

## Results

### Characteristics of participants

A total of 532 PLHIV (458 males and 74 females) aged 18–82 years were included in our study. Of these, 391 received the HAART regimen of 3TC+EFV+TDF at baseline, while 373 developed dyslipidemia during follow-up. The characteristics of the study participants, according to their dyslipidemia status, are shown in Table 1. Compared to the non-dyslipidemia participants, those with dyslipidemia were more likely to be older, have a higher mean level of BMI, and a history of diabetes. They also tended to have liver dysfunction, with higher TC, TG, and PLT levels at baseline.

### Prevalence of dyslipidemia

After HAART, the lipid profiles of PLHIV, including TC and TG, and the prevalence of dyslipidemia, further increased, as shown in the supporting information of S2 Fig. Table 2

**Table 1. Basic characteristics of participants.**

| Baseline variables | Non-dyslipidemia(N = 159) | Dyslipidemia(N = 373) | P-value |
|---|---|---|---|
| Age, years | 35.0(25.0,48.0) | 40.0(28.0,53.0) | 0.029 |
| Gender | | | 0.181 |
| Male | 132(83.0) | 326(87.4) | |
| Female | 27(17.0) | 47(12.6) | |
| BMI, kg/m$^2$ | 21.5±2.7 | 22.4±2.9 | 0.003 |
| Initial HAART regimens | | | 0.001 |
| 3TC+EFV+TDF | 132(83.0) | 259(69.4) | |
| 3TC+EFV+AZT | 27(17.0) | 114(30.6) | |
| Infection pathway | | | 0.160 |
| Heterosexual transmission | 92(57.9) | 245(65.7) | |
| MSM | 64(40.3) | 125(33.5) | |
| Others | 3(1.9) | 3(0.8) | |
| WHO clinical stage | | | 0.319 |
| I or II | 157(98.7) | 361(96.8) | |
| III or | 2(1.3) | 12(3.2) | |
| Education level | | | 0.261 |
| Illiterate or elementary school | 45(28.3) | 114(30.6) | |
| Junior or high or secondary school | 78(49.1) | 197(52.8) | |
| College and above | 36(22.6) | 62(16.6) | |
| With diabetes | | | 0.009 |
| No | 153(97.5) | 337(91.1) | |
| Yes | 4(2.5) | 33(8.9) | |
| CD4, cells/L | 270.0(180.0,400.0) | 285.8(180.3,435.0) | 0.379 |
| CD4/CD8 | 0.4(0.2,0.5) | 0.4(0.2,0.5) | 0.209 |
| TC, mmol/L | 3.8(3.4,4.2) | 4.3(3.8,4.8) | <0.001 |
| TG, mmol/L | 1.0(0.7,1.3) | 1.5(1.1,2.1) | <0.001 |
| WBC, 10$^9$/L | 5.3(4.2,6.6) | 5.3(4.3,6.4) | 0.725 |
| Platelet, 10$^9$/L | 178.0(148.0,218.0) | 195.0(159.0,234.0) | 0.007 |
| Hemoglobin, g/L | 146.0(133.0,154.0) | 149.0(134.0,158.0) | 0.153 |
| ALT, U/L | 21.0(15.0,30.0) | 22.2(15.7,33.0) | 0.098 |
| AST, U/L | 21.0(18.0,25.1) | 22.0(19.0,29.0) | 0.006 |
| Creatinine, mmol/L | 72.0(64.0,82.2) | 73.0(64.9,82.5) | 0.393 |
| TBIL, mmol/L | 10.5(6.8,14.6) | 9.8(7.3,13.6) | 0.420 |

**Abbreviations:**BMI: Body mass index; MSM: men who have sex with men; CD4: CD4[+] T-lymphocyte count; CD8: CD8[+] T-lymphocyte count; TC: total cholesterol; TG: triglycerides; WBC: White blood cell; ALT: Alanine aminotransferase; AST: Aspartate transaminase; TBIL: Total bilirubin; 3TC: lamivudine; EFV: efavirenz; TDF: tenofovir disoproxil fumarate; AZT: Zidovudine.

presented the prevalence of dyslipidemia among various subpopulations. The total prevalence of dyslipidemia reached 70.11%, with the prevalence of marginal elevated TC and marginal elevated TG being 36.28% and 62.78%, respectively. The prevalence of dyslipidemia was consistent across both sexes, but we observed a higher prevalence of marginal TG elevation in men compared to women. The prevalence of dyslipidemia generally increased with increasing BMI, particularly at the marginal elevated TG level. Participants with diabetes had a higher prevalence of dyslipidemia and abnormal lipid levels. The prevalence of dyslipidemia varied among different HAART regimens. In comparison to PLHIV who received 3TC+EFV+AZT, those receiving 3TC+EFV+TDF had a lower prevalence of dyslipidemia (80.9% vs 66.2%, *p*-

**Table 2. Prevalence of dyslipidemia by subpopulation.**

| Baseline variables | N | Dyslipidemia(%) | TC≥5.3mmol/L (%) | TG≥1.7mmol/L (%) |
|---|---|---|---|---|
| Total | 532 | 373(70.11) | 193(36.28) | 334(62.78) |
| Gender | | | | |
| Male | 458 | 326(71.2) | 163(35.6) | 298(65.1) |
| Female | 74 | 47(63.5) | 30(40.5) | 36(48.6) |
| | | | | ** |
| Age | | | | |
| 18–29 | 160 | 103(64.4) | 55(34.4) | 93(58.1) |
| 30–39 | 115 | 82(71.3) | 36(31.3) | 77(67.0) |
| 40–49 | 101 | 69(68.3) | 35(34.7) | 65(64.4) |
| ≥50 | 156 | 119(76.3) | 67(42.9) | 99(63.5) |
| BMI | | | | |
| <18.5 | 44 | 28(63.6) | 17(38.6) | 26(59.1) |
| 18.5–24 | 271 | 192(70.8) | 99(36.5) | 167(61.6) |
| ≥24 | 124 | 101(81.5) | 53(42.7) | 95(76.6) |
| | | * | | ** |
| Initial ART regimens | | | | |
| 3TC+TDF+EFV | 391 | 259(66.2) | 133(34.0) | 227(58.1) |
| 3TC+AZT+EFV | 141 | 114(80.9) | 60(42.6) | 107(75.9) |
| | | ** | | ** |
| With diabetes | | | | |
| Yes | 37 | 33(89.2) | 15(40.5) | 32(86.5) |
| No | 490 | 337(68.8) | 176(35.9) | 300(61.2) |
| | | ** | | ** |
| CD4 | | | | |
| <200 | 115 | 75(65.2) | 37(32.2) | 62(53.9) |
| 200–350 | 131 | 82(62.6) | 40(30.5) | 74(56.5) |
| ≥350 | 141 | 102(72.3) | 53(37.6) | 93(66.0) |
| CD4/CD8 | | | | |
| <0.4 | 215 | 137(63.7) | 64(29.8) | 124(57.7) |
| 0.4–1.0 | 152 | 110(72.4) | 61(40.1) | 95(62.5) |
| ≥1.0 | 20 | 12(60.0) | 5(25.0) | 10(50.0) |
| WHO clinical stage | | | | |
| I or II | 518 | 361(69.7) | 190(36.7) | 322(62.2) |
| III or | 14 | 12(85.7) | 3(21.4) | 12(85.7) |

*: P-value <0.05

**: P-value < 0.01

**Abbreviations:** BMI: Body mass index; CD4: CD4$^+$ T-lymphocyte count; CD8: CD8$^+$T-lymphocyte count; 3TC: lamivudine; EFV: efavirenz; TDF: tenofovir disoproxil fumarate; AZT: Zidovudine.

value < 0.01), and a lower prevalence of marginal elevated TG (75.9% vs 58.1%, *p*-value < 0.01).

## Evolution of dyslipidemia in PLHIV receiving different ART

486 PLHIV, none of them changed HAART during the follow-up period. The baseline rates of dyslipodemia, marginal elevated TC, and marginal elevated TG were similar in those receiving 3TC+EFV+AZT (dyslipodemia: 38.18%, TC ≥ 5.3 mmol/L: 11.82, TG ≥ 1.7mmol/L: 32.73%)

and those receiving 3TC+EFV+TDF (dyslipodemia: 37.77%, TC ≥ 5.3 mmol/L: 12.5,
TG ≥ 1.7mmol/L: 29.79%), as shown in the supporting information of S2 Table. After a pro-
longed period of HAART, the prevalence of dyslipodemia increased in these PLHIV, regard-
less of the regimen. However, the changes indyslipodemia varied among PLHIV receiving
different regimens, as evident in Fig 1. The rate of dyslipidemia in PLHIV receiving 3TC+EFV

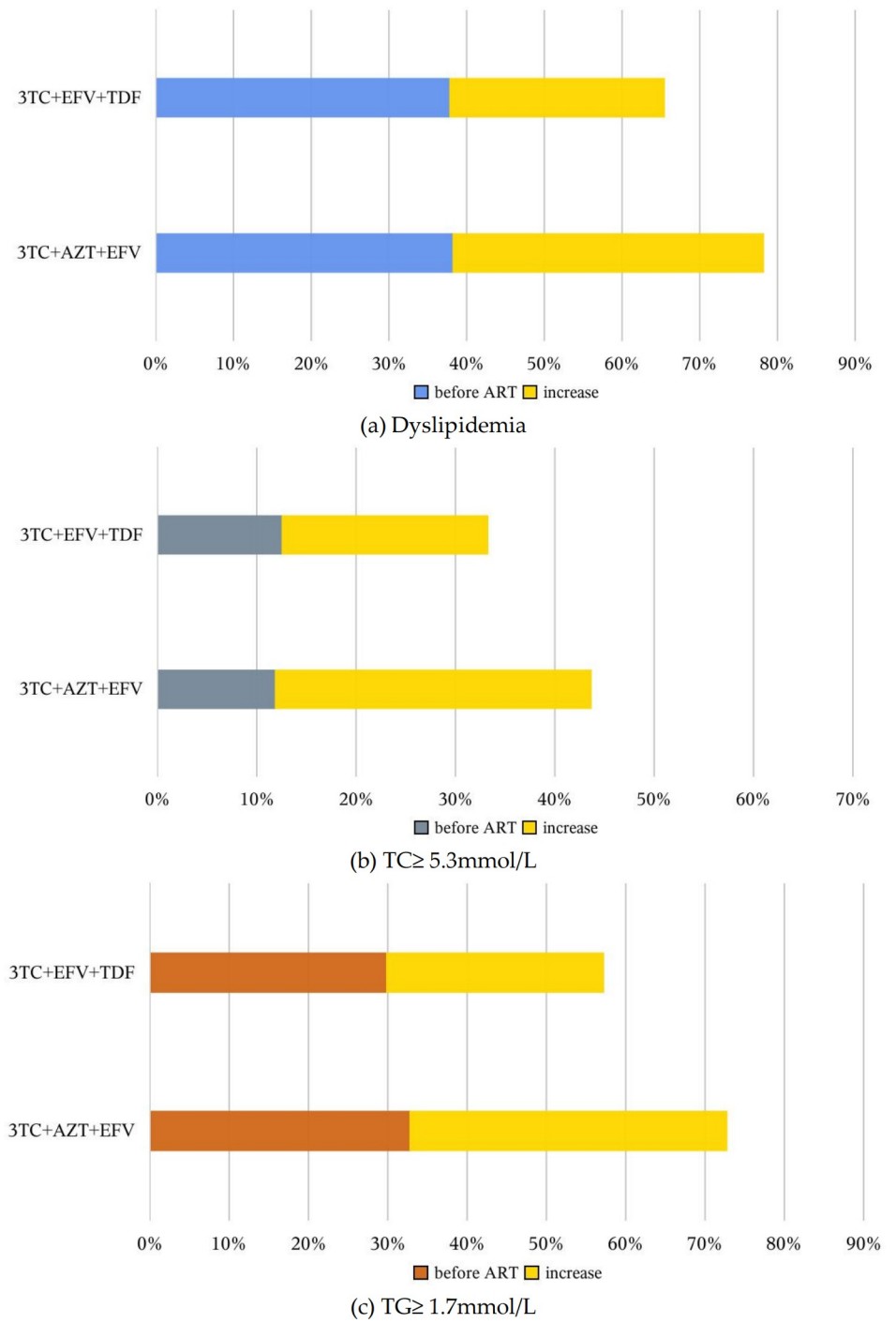

(a) Dyslipidemia

(b) TC≥ 5.3mmol/L

(c) TG≥ 1.7mmol/L

**Fig 1. Changes in the prevalence of dyslipidemia in PLHIV receiving different HAART regimens.**

**Table 3. Association of two initial HAART regimens with different lipid types.**

| Variables | N | #(%) | Crude | | Adjusted | |
|---|---|---|---|---|---|---|
| | | | OR (95%CI) | p-value | OR (95%CI) | p-value |
| Dyslipidemia | | | | | | |
| 3TC +EFV+TDF | 391 | 259(66.2) | Ref | Ref | Ref | Ref |
| 3TC +EFV+AZT | 141 | 114(80.9) | 2.15(1.35,3.44) | <0.01 | 2.09(1.28,3.41) | <0.01 |
| TC ≥ 5.3 | | | | | | |
| 3TC +EFV+TDF | 391 | 133(34.0) | Ref | Ref | Ref | Ref |
| 3TC +EFV+AZT | 141 | 60(42.6) | 1.44(0.97,2.13) | 0.07 | 1.39(0.93,2.09) | 0.11 |
| TG ≥ 1.7 | | | | | | |
| 3TC +EFV+TDF | 391 | 227(58.1) | Ref | Ref | Ref | Ref |
| 3TC +EFV+AZT | 141 | 107(75.9) | 2.27(1.47,3.51) | <0.01 | 2.40(1.50,3.84) | <0.01 |

**Note:** Dyslipidemia: adjusted for BMI, age, glu, cd4/cd8, alt, ast and plt; TC ≥ 5.3: adjusted for BMI, age, glu, route of transmission and education; TG ≥ 1.7: adjusted for bmi, age, sex, cd4, glu, scr, ast and platelet.

+AZT increased more dramatically, particularly in TC, with a percentage difference of 269.2%. Although the proportion of censoring at 3-year follow-up of these 486 patients was substantial, it is evident from the supporting information of S3 Fig that PLHIV who received 3TC+EFV +AZT consistently had a higher prevalence of dyslipodemia than those receiving the other regimen throughout the entire follow-up period. Within one year after HAART initiation, the prevalence of dyslipidemia in patients receiving 3TC+EFV+TDF even showed a gradual decreasing trend. The number of participants in the cohort at various follow-up points stratified by regimen is presented in the supporting information of S3 Table.

### Association of different combination antiretroviral therapies with dyslipidemia

Multivariate analysis revealed that the risk of dyslipidemia varied according to different HAART regimens. As shown in Table 3, the proportion of dyslipidemia in PLHIV receiving 3TC+EFV+AZT was obviously higher than in those receiving 3TC+EFV+TDF (80.9% vs 66.2%). The same trend was observed for marginal elevated TC (42.6% vs 34.0%) and marginal elevated TG (75.9% vs 58.1%). Compared to patients treated with 3TC+EFV+TDF, the risk of dyslipidemia for those under the treatment of a combination antiretroviral drug consisting of 3TC, EFV and AZT was significantly increased by 1.09 (OR = 2.09, 95%CI: 1.28–3.41), after adjusting for BMI, age, CD4/CD8, ALT, AST, platelet, and history of diabetes. Meanwhile, after adjusting for the corresponding covariates, theOR (95%CI) of an elevated TG level and an elevated TC level for those receiving 3TC+EFV+AZT were 2.40 (1.50–3.84) and 1.39 (0.93–2.09), respectively, when compared with patients receiving the other regimen. After considering the repeated measurements of blood lipids in the GEE model, PLHIV receiving 3TC+EFV +AZT consistently positive association with marginal elevated TG risk, with an OR value of 1.84 (95%CI: 1.10–3.07, the supporting information of S4 Table, the supporting information of S4 Fig). These results proved from multiple dimensions that the adverse effect on blood lipid metabolism of PLHIV who use the 3TC+EFV+AZT regimen is significantly greater than that of 3TC+EFV+TDF.

## Discussion

This retrospective cohort study is one of the few reports in China that focuses on evaluating the prevalence of dyslipidemia and the impact of various drug combinations on dyslipidemia

in PLHIV who are receiving the two most commonly used first-line free antiretroviral regimens.

The results of our study revealed that after antiretroviral treatment, the prevalence of dyslipidemia among HIV-infected individuals was 70.11%, which was significantly higher than the rate before HAART. Furthermore, the occurrence of increased TG levels was more common. The prevalence of hyperlipidemia among PLHIV across various studies ranges from 28% to 80%, with hypertriglyceridemia being the most common abnormality [9]. This wide range is understandable, given the intrinsic differences in study populations and the evolution of HIV drug treatments. A meta-analysis of 55 observational studies has reported that HIV-infected patients receiving antiretroviral therapy have significantly higher concentrations of total cholesterol and triglycerides compared to patients who never receive HAART [10], which is consistent with our findings.

Previous studies have reported that dyslipidemia in PLHIV is easily influenced by numerous factors. In addition to traditional factors such as age and genetic background, HIV infection-related chronic inflammation and abnormal immune activation, vascular endothelial cell dysfunction, and co-infection are also key factors affecting blood lipid metabolism [11–13]. Therefore, in order to comprehensively assess the actual impact of different HAART regimens on lipid profiles, we adjusted for various potential confounding factors.Our results clearly demonstrated that PLHIV receiving 3TC+EFV+AZT had a higher risk of dyslipidemia, particularly increased TG levels, compared to 3TC+EFV+TDF. This finding is broadly in line with two previous studies in Shenzhen, China,whichreported that 3TC+EFV+TDF has a lower risk of dyslipidemia than the other first-line free antiretroviral regimens available to PLHIV [14, 15].

Considering the limited drug resources, non-international first-line drugs, such as TDF, EFV and AZT, are still widely used in developing countries, as is the case in China. Moreover, the effects of various HAART drugs on lipid metabolism and related pathogenic mechanisms vary [16]. Sticprospective clinical studies have confirmed that the above-mentioned backbone drugs for free clinical application exhibit good virological and immunological effects [17]. In China, commonly used nucleoside reverse transcriptase inhibitors include 3TC, AZT, TDF, etc. Mitochondrial DNA replication ability and normal mitochondrial function in adipocytes can be inhibited by NRTIs, but the full mechanism of its effect on dyslipidemia remains unclear. Nucleoside drugs primarily cause lipoatrophy and hypertriglyceridemia [18]. AZT is more common. Interestingly, TDF has fewer adverse effects on lipid metabolism [19–21]. It can even activate the peroxisome proliferator-activated receptors expressed in the liver, upregulate the expression of CD36, and increase the uptake of free fatty acids in the circulation, which can reduce the concentration of TC, TG, LDL_C and Non_HDL_C in plasma. Commonly used non-nucleoside reverse transcriptase inhibitors include NVP, EFV and so on. They may affect blood lipid metabolism by regulating the expression of genes involved in cholesterol antiporters, but the specific pathogenesis remainsundetermined.Among them, EFV has a greater impact on blood lipids [22]. The ALTAIR study [23] showed that the use of EFV would cause a consistent increase in the concentration of different types of blood lipids. Nevertheless, since EFV was included in all HAART regimens in this study, its impact on outcomes was not considered. The above explanations support our hypothesis.

In HIV-infected patients, dyslipidemia is often observed and can significantly contribute to the elevated cardiovascular risk prevalent in this population [24]. Antiretroviral drug exposure appears to play a crucial role in this regard [25]. Consequently, optimizing the HAART regimen to enhance patient survival and prognosis is of considerable importance. Our findingsdemonstrated that 3TC+EFV+TDF is more suitable for managing blood lipids, provided there are no contraindications. Furthermore, a dynamic analysis of blood lipid metabolism reveals

that the rate of dyslipidemia significantly changesin the early stages of treatment. This rate of change slows down as thetreatment time extends, supporting the early selection of appropriate antiviral treatment regimens to enhance the feasibility of lipid metabolism.

This study is the first study in Huzhou on the independent effect of HAART regimen on dyslipidemia in PLHIV. We conducted a comprehensive analysis of the initial treatment regimens and dyslipidemia during the follow-up period, and established a baseline-characteristically matched longitudinal cohort. The double-dimensional design significantly enhanced the credibility of our findings. However, the study also had some limitations. Firstly, the lack of blood lipid information was a significant concern. The current study included only 532 PLHIV, and selection bias wasunavoidable. Nevertheless, we established strict inclusion and exclusion criteria to ensure the homogeneity of participants as much as possible. Secondly, the absence of high-density lipoprotein cholesterol and low-density lipoprotein cholesterol in our database was a limitation. Tocircumventunderestimating the prevalence of dyslipidemia, we used the cut-off value of marginal elevated blood lipids to determine lipid status. Furthermore, when conducting longitudinal analysis, we reduced the sample size duetothe lack of regular blood testing for all patients. Despitethe loss to follow-up occurring at different time points, we observed significant differences between the two HAART regimens. Additionally, we lacked information on the use of lipid-modifying agents, which could have influenced our results to a certain extent.

In the future, we strongly recommend that staff incorporate all critical lipid profiles into the health system and diligently prepare to conduct prospective cohort studies. These studies will evaluate longitudinal changes in these parameters in PLHIV receiving various HAART regimens. The laws of blood lipid metabolism in HIV-infected patients will be more accurately described, and the association between different HAART regimens and dyslipidemia will also be assessed in greater detail.

## Conclusions

The prevalence of blood lipid abnormalities varied among different classes of drugs. When compared to 3TC+EFV+AZT, 3TC+EFV+TDF have a lower risk of dyslipidemia. For patients who are potentiallyatrisk for cardiovascular disease, we recommend that when receiving HAART in the early stages, they should choose a HAART regimen that is beneficial for blood lipids.

## Supporting information

**S1 Fig. The study workflow.**
(TIF)

**S2 Fig. Changes of blood lipid and prevalence of dyslipidemia in PLHIV after HAART.** TG and TC with obviously skewed distribution, they are expressed logarithmically.
(TIF)

**S3 Fig. Evolution of dyslipidemia in PLHIV receiving different HAART regimens during 3 years of follow-up.**
(TIF)

**S4 Fig. Repeat measurements of blood lipids in PLHIV receiving different HAART regimens.**
(TIF)

**S1 Table. Comparison of pre- and post-imputations by 5-fold multiple imputation.** Continuous variables was described as median (1$^{st}$ quartile, 3$^{rd}$ quartile) as its distribution was skewed and Mann-Whitney U test was applied to compare the difference between two groups; Categorical data were presented with number (%) and chi-square tests or Fisher's exact test were used to compare the differences between pre- and post-imputations data. **Abbreviations:** CD4: CD4$^+$ T-lymphocyte count; CD8: CD8$^+$ T-lymphocyte count; WBC: White blood cell; ALT: Alanine aminotransferase; AST: Aspartate transaminase; TBIL: Total bilirubin; FPG: Fast plasma glucose.
(DOCX)

**S2 Table. Changes in the prevalence of dyslipidemia in PLHIV receiving different HAART regimens.** Percentage of change = (prevalence of dyslipidemia after HAART- prevalence of dyslipidemia before HAART)/ prevalence of dyslipidemia before HAART.
(DOCX)

**S3 Table. Numbers of patients in the cohort at various follow-up points stratified by regimens.**
(DOCX)

**S4 Table. The gee model of effect of two HAART regimens on the risk of dyslipidemia of PLHIV.** 77 pairs of PLHIV were included the gee model, in which one case (PLHIV receiving 3TC+EFV+AZT) was matched by age, BMI, cd4, cd8, TG and TC at baseline with one control (PLHIV receiving 3TC+EFV+TDF).
(DOCX)

## Acknowledgments

The authors would like to thank all participants for their valuable contributions to this study.

## Author Contributions

**Conceptualization:** Yanan Wang, Zhongrong Yang, Jing Li, Xiaoqi Liu, Feilin Ren, Meihua Jin, Guangyun Mao.

**Data curation:** Yanan Wang, Zhenqian Wu, Hui Wang, Yuxin Chen, Ziyi Wang, Zhaowei Tong, Xiaofeng Li, Guangyun Mao.

**Formal analysis:** Yanan Wang, Hui Wang, Yuxin Chen, Ziyi Wang, Guangyun Mao.

**Funding acquisition:** Zhongrong Yang, Jing Li, Xiaoqi Liu, Feilin Ren, Meihua Jin.

**Investigation:** Zhenqian Wu, Zhaowei Tong, Xiaofeng Li.

**Methodology:** Yanan Wang, Hui Wang, Yuxin Chen, Ziyi Wang, Guangyun Mao.

**Supervision:** Zhongrong Yang, Jing Li, Xiaoqi Liu, Feilin Ren, Meihua Jin, Guangyun Mao.

**Writing – original draft:** Yanan Wang.

**Writing – review & editing:** Zhongrong Yang, Jing Li, Xiaoqi Liu, Feilin Ren, Meihua Jin, Guangyun Mao.

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
