## [Decision Letter · Decision Letter 0]

10 Aug 2023

PONE-D-23-04082The Risk of Dyslipidemia on PLHIV Associated with Different Antiretroviral Regimens in HuzhouPLOS ONE

Dear Dr. 王,

Thank you for submitting your manuscript to PLOS ONE. After careful consideration, we feel that it has merit but does not fully meet PLOS ONE’s publication criteria as it currently stands. Therefore, we invite you to submit a revised version of the manuscript that addresses the points raised during the review process.

We look forward to receiving your revised manuscript.

Kind regards,

Manuela Ceccarelli

Academic Editor

PLOS ONE

Journal Requirements:

The authors declare that the research was conducted in the absence of any commercial or financial relationships that could be construed as a potential conflict of interest.

Additional Editor Comments:

Dear authors,

the reviewers read with interest your manuscript and expressed their opinions. My decision, on the basis of their suggestions, is "major revisions".

Please revise your manuscript according to the reviewers' observations.

Kind regards.

Reviewers' comments:

Reviewer's Responses to Questions

**Comments to the Author**

1. Is the manuscript technically sound, and do the data support the conclusions?

Reviewer #1: Yes

Reviewer #2: Yes

2. Has the statistical analysis been performed appropriately and rigorously? 

Reviewer #1: No

Reviewer #2: Yes

3. Have the authors made all data underlying the findings in their manuscript fully available?

Reviewer #1: Yes

Reviewer #2: Yes

4. Is the manuscript presented in an intelligible fashion and written in standard English?

Reviewer #1: Yes

Reviewer #2: Yes

5. Review Comments to the Author

Reviewer #1: I read this manuscript with great interest and want to acknowledge the authors for their excellent work. My main concern in this manuscript is data management.

I’m wondering about the data analysis of this study is logistic regression. I expected a time-to-event analysis or joint models since the data has a time component, repeated measurement of outcome variables (lipid profiles), and comparing the two HARRT regimens (3TC-EFV-TDF and 3TC-EFV-AZT). The epidemiological measures for cohort study with an open population are incidence density and measures of association again be relative risk or hazard ratio, but this paper reported prevalence rate and odds ratio for retrospective cohort study. I will be happy if the authors of this manuscript address these issues.

Reviewer #2: 1) There are some typos in the lines 203-214 where dyslipidemia is written as "dyslipodemia", please correct it.

2) In line 32, method section of abstract, the authors wrote "from June 2010 to June 2022", while in all the other section that period seems to be from July 2005 to June 2022 (lines 80-81 and 91), please correct it;

3) Data about the use of lowering triglycerides/cholesterol drugs are missing in the analysis. It would be interesting to know if the reduction in lipid profile was observed in people treated or not with those drugs. Could the authors provide this information about the participants, possibly making a sub-analysis that include this variable?

6. PLOS authors have the option to publish the peer review history of their article (what does this mean?). If published, this will include your full peer review and any attached files.

Reviewer #1: No

Reviewer #2: No

---

## [Author Response · Author response to Decision Letter 0]

4 Sep 2023

COMMENTS TO THE AUTHOR:

Reviewer#1:

Question1. I’m wondering about the data analysis of this study is logistic regression. I expected a time-to-event analysis or joint models since the data has a time component, repeated measurement of outcome variables (lipid profiles), and comparing the two HARRT regimens (3TC-EFV-TDF and 3TC-EFV-AZT). The epidemiological measures for cohort study with an open population are incidence density and measures of association again be relative risk or hazard ratio, but this paper reported prevalence rate and odds ratio for retrospective cohort study. I will be happy if the authors of this manuscript address these issues.

Response: Thank you very much for this valuable comment. Our primary statistical method of choice is logistic regression model, rather than the Cox proportional hazards model. The reason for this is that our outcome of interest is dyslipidemia rather than death. Although various laboratory measurements, including triglyceride levels, were periodically assessed during the patient’s follow-up, we do not have access to the exact timing of when lipid abnormalities occurred. In other words, we are unable to obtain the true survival time of the patients. Therefore, our outcome is defined as the occurrence of elevated lipid profiles at any point during the follow-up period, which does not allow for Cox regression analysis. Additionally, to avoid wasting the repeated measurements taken during the follow-up, we employed the generalized estimating equation (GEE) model in the second analysis to evaluate the association between treatment regimen and lipid abnormalities among patients who did not change their treatment during the follow-up. Moreover, this study is a retrospective cohort study, and retrospective studies cannot calculate incidence rates. Since the relative risk (RR) is based on rates, the measure of association in this article is presented as odds ratio (OR) values.

Reviewer#2:

Question1. There are some typos in the lines 203-214 where dyslipidemia is written as "dyslipodemia", please correct it.

Response: We are grateful for this kind reminder and sorry for this carelessness. We have completed the modification of “dyslipidemia”.

Question2. In line 32, method section of abstract, the authors wrote "from June 2010 to June 2022", while in all the other section that period seems to be from July 2005 to June 2022 (lines 80-81 and 91), please correct it

Response: Thank you again for your kind reminder and we are sorry for our carelessness. We have corrected the error in line 32.

Question3. Data about the use of lowering triglycerides/cholesterol drugs are missing in the analysis. It would be interesting to know if the reduction in lipid profile was observed in people treated or not with those drugs. Could the authors provide this information about the participants, possibly making a sub-analysis that include this variable?

Response: Many thanks for this important and interesting reminder. We completely agree with the reviewer. Unfortunately, there is no information in our data set on drug use other than antiviral regimens. We believe that if we obtain the relevant variables of lipid-lowering drugs, we may get some new results. In subsequent studies, we will recommend that CDC staff collect additional information about patients' drug use.

---

## [Decision Letter · Decision Letter 1]

22 Feb 2024

PONE-D-23-04082R1The risk of dyslipidemia associated with different antiretroviral therapy regimens in people living with HIV in Huzhou,ChinaPLOS ONE

Dear Dr. 王,

Thank you for submitting your manuscript to PLOS ONE. After careful consideration, we feel that it has merit but does not fully meet PLOS ONE’s publication criteria as it currently stands. Therefore, we invite you to submit a revised version of the manuscript that addresses the points raised during the review process.

.

We look forward to receiving your revised manuscript.

Kind regards,

Cavin Epie Bekolo, MD, MSc

Academic Editor

PLOS ONE

Reviewers' comments:

Reviewer's Responses to Questions

**Comments to the Author**

1. If the authors have adequately addressed your comments raised in a previous round of review and you feel that this manuscript is now acceptable for publication, you may indicate that here to bypass the “Comments to the Author” section, enter your conflict of interest statement in the “Confidential to Editor” section, and submit your "Accept" recommendation.

Reviewer #1: (No Response)

Reviewer #3: (No Response)

Reviewer #4: All comments have been addressed

2. Is the manuscript technically sound, and do the data support the conclusions?

Reviewer #1: Yes

Reviewer #3: Yes

Reviewer #4: Partly

3. Has the statistical analysis been performed appropriately and rigorously? 

Reviewer #1: No

Reviewer #3: Yes

Reviewer #4: I Don't Know

4. Have the authors made all data underlying the findings in their manuscript fully available?

Reviewer #1: Yes

Reviewer #3: No

Reviewer #4: Yes

5. Is the manuscript presented in an intelligible fashion and written in standard English?

Reviewer #1: Yes

Reviewer #3: Yes

Reviewer #4: No

6. Review Comments to the Author

Reviewer #1: Sorry for this; the method of data analysis for this study doesn’t convince me. It is clear that survival analysis can apply to any outcome with a time component, not just death. For example, patients with longer follow-up times have a higher chance of developing dyslipidaemia than patients with shorter follow-up times. So, the analysis should consider follow-up time accounting interval censoring; otherwise, the result will be misleading. In the GEE analysis, the authors assessed those with unchanged HAART regimen, here what if the change is because of high lipid profile, does it mean you introducing bias in the analysis. Related question: how did you manage patients dead or lost in the follow-up period and having one or more lipid measurements? Did you assess the computing risks?

Reviewer #3: The authors have approached the subject in an easily understandable way. However, has the possibility of interference with other products taken by patients been taken into account? This could have been a factor influencing the advent of elevation of lipid profiles, can the authors give more details on this subject.

Reviewer #4: Dear Authors,

Thank you for the opportunity to review your manuscript. Your study addresses an important area of research, and I appreciate your efforts in exploring this subject. I have several comments and suggestions that I believe could enhance the clarity, accuracy, and impact of your paper:

1. I understand that English might not be the first language of the authors, but they might want to seek professional services to improve the presentation and grammar. This can remarkably improve the readability and professional presentation of your manuscript.

2. Research Question and Aims: The research question and specific aims of the study are not stated with sufficient clarity. Clearly articulating these elements is fundamental to guiding the reader through your research process and understanding the significance of your findings. I recommend revising the introduction to explicitly state the research question(s)/aims. For example, to estimate the prevalence of...., to assess the trend.... to evaluate the association of ART with dyslipedemia. When clearly stated readers expect to see results and discussion sections presented in alignment with the study aims. The current presentation is hard to follow.

3.Lines 269-272: The passage discussing the comparative adverse effects of different ART regimens on blood lipid metabolism seems more interpretative than is typical for a Results section. I recommend relocating or rephrasing this content to maintain a clear distinction between the presentation of findings (Results) and their interpretation (Discussion).

4. Including both unadjusted and adjusted regression analyses results for the covariates would offer a more comprehensive view of the data and the factors influencing dyslipidemia among PLHIV. Presenting the effect sizes of other potential determinants would also be beneficial in interpreting the results and understanding their broader implications. You could add it as supplementary data.

I believe that addressing these points will greatly improve the manuscript's quality and contribution to the field.

7. PLOS authors have the option to publish the peer review history of their article (what does this mean?). If published, this will include your full peer review and any attached files.

Reviewer #1: No

Reviewer #3: No

Reviewer #4: No

---

## [Author Response · Author response to Decision Letter 1]

21 Apr 2024

Reviewer#1:

Question. It is clear that survival analysis can apply to any outcome with a time component, not just death. For example, patients with longer follow-up times have a higher chance of developing dyslipidaemia than patients with shorter follow-up times. So, the analysis should consider follow-up time accounting interval censoring; otherwise, the result will be misleading. In the GEE analysis, the authors assessed those with unchanged HAART regimen, here what if the change is because of high lipid profile, does it mean you introducing bias in the analysis. Related question: how did you manage patients dead or lost in the follow-up period and having one or more lipid measurements? Did you assess the computing risks?

Response: Thank you very much for your valuable comments. It is well known that survival analysis can be utilized for outcomes with a time component, provided that we have a relatively accurate survival time, that is, between the start of the study and the occurrence of the event of interest. Death is just one example we provide, as we can usually ascertain the actual time of death. In clinical practice, the time of visit is often used as a proxy for the time of occurrence of a disease, despite the potential for measurement errors. However, the survival analysis method is robust to this situation in the data. But I would like to say the outcomes of our study differ from those mentioned above. Lipids were measured at each follow-up, and as long as a patient had dyslipidemia, we considered that the end event of interest had occurred. Furthermore, our lipid data are incomplete. It is likely that patients already had dyslipidemia at the first follow-up, but were missed due to a lack of HDL_C or LDL_C data. These individuals were either not identified until later follow-ups based on other lipid data, or remained undetected. Consequently, the outcome was more suitable for analysis as a binary event rather than as an event that included time. In summary, the case is specific. Generally, survival analysis methods should be selected for data containing time, but for this study, we believe that logistic regression analysis is more suitable.

In the GEE analysis, we assessed those with unchanged HAART regimen. First of all, we cannot deny that patients may have some adverse reactions during treatment, such as metabolic disorders, and change the HAART regimen, which will cause bias in the study results. However, we were unable to ascertain the specific reasons for patients' treatment regimen changes, thereby precluding an estimation of this bias. This constitutes a significant limitation of our study. Nonetheless, 89.5% (476/532) of patients maintained their treatment regimen, suggesting a minimal bias. Secondly, the GEE analysis necessitated repeated measurements, hence we retained patients with at least two follow-up records.

Considering the interference caused by confounding factors such as age and CD4, the relationship between HAART and dyslipidemia was analyzed after 1:1 matching. Ultimately, GEE analysis was performed on only 77 pairs of patients after receiving matching, and none of these 77 pairs disappeared or died during follow-up. What I would like to emphasize is that, despite the fact that only 154 subjects remain after pairing, GEE analysis serves as an auxiliary analysis supporting the main research findings in this study. This part of the results is also included in a supplementary document.Thank you again for your comments.

Reviewer#3:

Question. The authors have approached the subject in an easily understandable way. However, has the possibility of interference with other products taken by patients been taken into account? This could have been a factor influencing the advent of elevation of lipid profiles, can the authors give more details on this subject.

Response: We are grateful for this kind reminder and we completely agree with your opinion. We must acknowledge that if patients take other medications, it may affect their blood lipid levels. Unfortunately, details of other drug use histories were missing from our database, which we will add to the limitations of the study.In subsequent studies, we will recommend that CDC staff collect additional information about patients' drug use.

Reviewer#4:

Question1. I understand that English might not be the first language of the authors, but they might want to seek professional services to improve the presentation and grammar. This can remarkably improve the readability and professional presentation of your manuscript.

Response: Thank you for your kind reminder and we will continue to polish our article to improve its readability.

Question2. Research Question and Aims: The research question and specific aims of the study are not stated with sufficient clarity. Clearly articulating these elements is fundamental to guiding the reader through your research process and understanding the significance of your findings. I recommend revising the introduction to explicitly state the research question(s)/aims. For example, to estimate the prevalence of...., to assess the trend.... to evaluate the association of ART with dyslipedemia. When clearly stated readers expect to see results and discussion sections presented in alignment with the study aims. The current presentation is hard to follow.

Response: We are grateful for this kind reminder. We have revised the research purpose of the introduction section to make it clearer and easier to understand.

Question3. Lines 269-272: The passage discussing the comparative adverse effects of different ART regimens on blood lipid metabolism seems more interpretative than is typical for a Results section. I recommend relocating or rephrasing this content to maintain a clear distinction between the presentation of findings (Results) and their interpretation (Discussion).

Response: Many thanks for this important and interesting reminder. We have deleted and modified the statements in lines 269-272 to make the description more logical.

Question4. Including both unadjusted and adjusted regression analyses results for the covariates would offer a more comprehensive view of the data and the factors influencing dyslipidemia among PLHIV. Presenting the effect sizes of other potential determinants would also be beneficial in interpreting the results and understanding their broader implications. You could add it as supplementary data.

Response: Many thanks for this kind reminder. However, we would like to say that this study is an association study, not an analysis of influencing factors. We had only one main independent variable, the HAART regimen, and the other variables in the multifactor model were corrected as confounders. And the effect sizes of HAART's effects on dyslipidemia, whether uncorrected or corrected, have been sorted out and placed in Table 3 of the paper. We do not think it is necessary to label the effect sizes of other confounding factors as this is not in line with the main idea of the article. Thank you again for your comments.

---

## [Decision Letter · Decision Letter 2]

31 May 2024

The Risk of Dyslipidemia on PLHIV Associated with Different Antiretroviral Regimens in Huzhou

PONE-D-23-04082R2

Dear Dr. 王,

We’re pleased to inform you that your manuscript has been judged scientifically suitable for publication and will be formally accepted for publication once it meets all outstanding technical requirements.

Kind regards,

Cavin Epie Bekolo, MD, MSc

Academic Editor

PLOS ONE

Additional Editor Comments (optional):

Reviewers' comments:

Reviewer's Responses to Questions

**Comments to the Author**

1. If the authors have adequately addressed your comments raised in a previous round of review and you feel that this manuscript is now acceptable for publication, you may indicate that here to bypass the “Comments to the Author” section, enter your conflict of interest statement in the “Confidential to Editor” section, and submit your "Accept" recommendation.

Reviewer #1: All comments have been addressed

Reviewer #3: All comments have been addressed

2. Is the manuscript technically sound, and do the data support the conclusions?

Reviewer #1: (No Response)

Reviewer #3: Yes

3. Has the statistical analysis been performed appropriately and rigorously? 

Reviewer #1: Yes

Reviewer #3: Yes

4. Have the authors made all data underlying the findings in their manuscript fully available?

Reviewer #1: Yes

Reviewer #3: No

5. Is the manuscript presented in an intelligible fashion and written in standard English?

Reviewer #1: Yes

Reviewer #3: Yes

6. Review Comments to the Author

Reviewer #1: (No Response)

Reviewer #3: (No Response)

7. PLOS authors have the option to publish the peer review history of their article (what does this mean?). If published, this will include your full peer review and any attached files.

Reviewer #1: No

Reviewer #3: No

---

## [Editor Report · Acceptance letter]

6 Aug 2024

PONE-D-23-04082R2 

PLOS ONE

Dear Dr. 王, 

I'm pleased to inform you that your manuscript has been deemed suitable for publication in PLOS ONE. Congratulations! Your manuscript is now being handed over to our production team.

Kind regards, 

on behalf of

Dr. Cavin Epie Bekolo 

Academic Editor

PLOS ONE